

# Cordycepin inhibits LPS-induced inflammatory and matrix degradation in the intervertebral disc

Yan Li[1,*], Kang Li[1,*], Lu Mao[2], Xiuguo Han[1], Kai Zhang[1], Changqing Zhao[1] and Jie Zhao[1]

[1] Shanghai Key Laboratory of Orthopedic Implants, Department of Orthopedics, Shanghai Ninth People's Hospital, Shanghai JiaoTong University School of Medicine, Shanghai, China
[2] Spine Center, Zhongda Hospital, School of Medicine, Southeast University, Nanjing, China
[*] These authors contributed equally to this work.

## ABSTRACT

Cordycepin is a component of the extract obtained from *Cordyceps militaris* and has many biological activities, including anti-cancer, anti-metastatic and anti-inflammatory effects. Intervertebral disc degeneration (IDD) is a degenerative disease that is closely related to the inflammation of nucleus pulposus (NP) cells. The effect of cordycepin on NP cells in relation to inflammation and degeneration has not yet been studied. In our study, we used a rat NP cell culture and an intervertebral disc (IVD) organ culture model to examine the inhibitory effects of cordycepin on lipopolysaccharide (LPS)-induced gene expression and the production of matrix degradation enzymes (MMP-3, MMP-13, ADAMTS-4, and ADAMTS-5) and oxidative stress-associated factors (nitric oxide and PGE2). We found a protective effect of cordycepin on NP cells and IVDs against LPS-induced matrix degradation and macrophage infiltration. In addition, western blot and luciferase assay results demonstrated that pretreatment with cordycepin significantly suppressed the LPS-induced activation of the NF-$\kappa$B pathway. Taken together, the results of our research suggest that cordycepin could exert anti-inflammatory and anti-degenerative effects on NP cells and IVDs by inhibiting the activation of the NF-$\kappa$B pathway. Therefore, cordycepin may be a potential treatment for IDD in the future.

Corresponding author
Jie Zhao, medzhaojie@163.com

## INTRODUCTION

Intervertebral disc degeneration (IDD) is thought to be a significant contributor to the development of low back pain (LBP) (*Kuslich, Ulstrom & Michael, 1991*; *Schwarzer et al., 1994*). In addition, LBP is one of the most common musculoskeletal complaints, estimated to reach up to 2.8% of healthcare visits in the United States (*Hart, Deyo & Cherkin, 1995*). The mechanism of IDD contains a complex biochemical cascade. One of the most important features of IDD is the loss of proteoglycan (PG) content of intervertebral discs (IVDs). This process is closely related to the gene expression and activity of disintegrins and metalloproteinases with thrombospondin motifs (ADAMTSs), matrix metalloproteinases (MMPs) and tissue inhibitors of metalloproteinases (TIMPs) (*Patel et al., 2007*; *Bachmeier et al., 2009*). As a strong promoter of inflammation, previous studies have shown that

lipopolysaccharide (LPS) can induce gene upregulation and the production of various proinflammatory cytokines and matrix-degrading enzymes, including MMP-3, MMP-13, ADAMTS-4 and ADAMTS-5 in NP cells, thus causing a reduction in PG content and IDD (*Ellman et al., 2012*; *Iwata et al., 2013*). Moreover, proinflammatory cytokine such as interleukin-1 (IL-1$\beta$) and tissue necrosis factor-$\alpha$ (TNF-$\alpha$) also play important roles in IDD (*Wuertz & Haglund, 2013*). Cytokines do not directly degrade the IVD like MMPs or ADAMTSs do; instead, they accelerate IDD by promoting the production of inflammatory substances by the disc cells (*Kepler et al., 2013*). As a TLR ligand, LPS can also initiate TLR signaling in NP cells, leading to the increased expression of proinflammatory cytokines and MMPs (*Klawitter et al., 2014*; *Rajan et al., 2013*). In addition to MMPs and ADAMTSs, cytokines can also induce chemokine ligand (CCL) expression in NP cells (*Wang et al., 2013*). According to previous research, CCLs can promote macrophage migration into the IVD, exacerbating the inflammatory state and causing pain (*Wang et al., 2013*; *You et al., 2013*).

*Cordyceps militaris* is a traditional Chinese medicine that has been widely used for decades. Recent studies have demonstrated that bioactive components isolated from Cordyceps species have various pharmacological functions (*Yue et al., 2013*; *Paterson, 2008*). Cordycepin (3′-deoxyadenosine) is one of the most widely studied components of *C. militaris* and has diverse bioactivities, such as anti-cancer, anti-metastatic and anti-inflammatory effects (*Nakamura, Shinozuka & Yoshikawa, 2015*; *Lee, Kim & Moon, 2010*; *Jeong et al., 2010*). Many studies have shown that anti-inflammatory treatments is an effective therapy for treating IDD *in vitro* (*Yang et al., 2015*; *Walter et al., 2015*). However, the results may be quite different between *in vitro* and *in vivo*, and the precise delivery of therapeutic agents to IVDs without causing tissue damage and while still maintaining an appropriate concentration for a long time is very difficult. To address some aspects of this problem, the present study used both *in vitro* and ex vivo models to investigate the inhibitory effect of cordycepin on LPS-induced inflammation and matrix decrease in intervertebral discs. Based on our findings, we suggest that cordycepin may be a potential treatment for IDD in the future.

## MATERIALS AND METHODS

### Reagents and animal ethics

Cordycepin, LPS and dimethylmethylene blue (DMMB) were purchased from Sigma (St. Louis, MO, USA). Cordycepin was dissolved in DMSO; the final concentration of DMSO in the medium was less than 0.05%. The same volume of DMSO was added to the control and LPS groups in all experiments. The NF-$\kappa$B luciferase reporter and the pRL-TK plasmids were purchased from Promega (Madison, WI, USA). Sprague Dawley rats were euthanized via the abdominal injection a lethal dose of pentobarbital sodium. All of the animal work was conducted according to relevant national and international guidelines and was approved by the Animal Experimental Ethical Committee of Shanghai Ninth People's Hospital (Approval number: 2013-47).

## NP cell isolation and culture

Nucleus pulposus (NP) cells were isolated from the lumbar spines of Sprague Dawley rats (6–8 weeks old, mixed male and female). The spines was separated between each of the lumbar discs, and then, a sterile scalpel blade was used to completely remove the nucleus pulposus. Before digestion with trypsin and collagenase, the nucleus pulposus was washed with PBS to remove other cells that may have been attached to the surface of nucleus pulposus. NP cells were cultured in complete medium (high-glucose DMEM with 10% FBS, 100 U/ml penicillin and 100 µg/ml streptomycin) up to passage 2–3.

## Cell viability assay

A commercial kit (Cell Counting Kit-8, CCK-8; Dojindo) was used to evaluate the potential cytotoxic effect of cordycepin. NP cells were plated in 96-well plates at a density of $5 \times 10^3$ per well, incubated with various concentrations of cordycepin for 24 h, administered 10 µl of CCK-8 solution and incubated for an additional 2 h. The optical density (OD) of each well was measured at 450 nm. The culture medium was used as a blank. Cell viability was calculated as follows: cell viability = [OD (with cordycepin) − OD (blank)]/[OD (without cordycepin) − OD (blank)].

## ELISA assessments

NP cells were incubated with various concentrations of cordycepin for 2 h and then stimulated with 10 µg/ml LPS for 24 h. The PGE2, MMP-3 and MMP-13 levels in the culture medium were measured using commercially available enzyme-linked immunosorbent assay kits according to the manufacturer's instructions (R&D Systems).

## Measurement of nitric oxide

Nitric oxide (NO) was measured by Griess reagent (*Imamura et al., 2015*). Briefly, NP cells were incubated with 100 µM cordycepin for 2 h and then stimulated with 10 µg/ml LPS (Sigma) for 24 h. Then, 50 µl of each culture supernatant was incubated at room temperature for 15 min with 50 µl of Griess reagent (Sigma) in a 96-well plate. The absorbance at 540 nm was measured. A standard curve was made using $NaNO_2$ to calculate the NO concentration of each sample.

## Immunofluorescence

NP cells were stimulated with 10 µg/ml LPS in the presence or absence of cordycepin (100 µM) in a 24-well plate for 5 days. Then, the cells were fixed with 4% paraformaldehyde for 30 min, treated with 0.1% Triton X-100 for 10 min, and blocked with 2% bovine serum albumin (Sigma) for 1 h. After being washed by PBS, the NP cells were incubated with an anti-collagen II antibody (1:50; Cell Signaling Technology) overnight at 4 °C and then exposed to Alexa Fluor® 594-conjugated secondary antibodies (1:100 dilution; Life Technologies) for 60 min at room temperature. Then, after being washed by PBS, the NP cells were counterstained with DAPI and phalloidin. Laser confocal microscopy (OLYMPUS) was used for observation and imaging. The integral optical density (IOD) of each picture was measured using the Image-Pro Plus 6.0 software (Media Cybernetics). The immunofluorescence results are expressed as IOD /cell number per view.

## Cell migration assay

A total of $1 \times 10^5$ RAW 264.7 cells were seeded on a Matrigel-coated polycarbonate membrane insert (8.0 µm pores) in a transwell apparatus (Costar) and maintained in 100 µl of complete medium (high-glucose DMEM with 10% FBS and 1% antibiotic). NP cells were also cultured in complete medium with or without 10 µg/ml LPS or 100 µM cordycepin in the lower chamber for 24 h. Then, the inserts were washed with PBS, and the cells on the top surface of the insert were carefully removed by using a cotton swab. The cells on the bottom surface of the insert were fixed with 4% paraformaldehyde for 10 min, followed by staining with 0.1% crystal violet for 20 min, and then subjected to an inspection and cell count via microscopy. The cells were counted under $200\times$ magnification, and the counts of 5 randomly chosen fields were averaged for each sample.

## RNA isolation and PCR

NP cells were incubated with various concentrations of cordycepin for 2 h and then stimulated with 10 µg/ml LPS for 24 h. Then, the total RNA of the NP cells was isolated using TRIzol reagent (Invitrogen) following the manufacturer's instructions. Reverse transcription was carried out from 1 µg of RNA using the 1st Strand cDNA Synthesis Kit (TAKARA) for first-strand complementary DNA (cDNA) synthesis. The relative gene expression was determined by real-time PCR. Real-time PCR was performed using the SYBR Premix Ex Taq kit (TAKARA) with the ABI Prism 7500 Fast Real-Time PCR system (Applied Biosystems) according to the manufacturer's instructions. The primers were designed and selected using BLAST. Gene expression was measured using the $2^{-\Delta\Delta Ct}$ method (*Schmittgen & Livak, 2008*). The primer sequences are summarized in Table 1 and we used $\beta$-actin as the internal control.

## Western blotting

For pathway related protein assays, NP cells were pretreated with various concentrations of cordycepin for 2 h and then stimulated with 10 µg/ml LPS for 30 min. For collagen II and aggrecan protein assays, NP cells were cultured with various concentrations of cordycepin and 10 µg/ml LPS for 5 days. Then, all cells were washed twice with cold PBS, and the total protein was extracted using an RIPA lysis buffer. The protein concentration was quantified using a BCA Protein Assay Kit (Thermo Scientific), Samples (20 µg of protein) were loaded into gel, separated by 10% SDS-PAGE and transferred onto polyvinylidene fluoride (PVDF) membranes (Millipore). The transfer membranes were blocked with 5% fat-free milk at room temperature for 1 h and then incubated with primary antibodies against ERK1/2, p-ERK1/2, JNK, p-JNK, p38, p-p38, I$\kappa$B$\alpha$, p-I$\kappa$B$\alpha$, p65, p-p65 (1:1,000, Cell Signaling Technology) aggrecan and collagen-II (1:1,000, Abcam) at 4 °C overnight. After three washes with TBST (TBS with Tween20), the membranes were incubated with appropriate secondary antibodies that were conjugated with IRDye 800CW at room temperature for 1 h. Immunoreactive bands were detected using the Odyssey infrared imaging system (LI-COR). The $\beta$-actin antibody (1:2,000; Cell Signaling Technology) was used as a control.

**Table 1  Sequences of the primers used in the polymerase chain reaction (PCR).**

| Gene | | Primer sequences (5′–3′) |
| --- | --- | --- |
| MMP-3 | Forward | TTTGGCCGTCTCTTCCATCC |
| | Reverse | GCATCGATCTTCTGGACGGT |
| MMP-13 | Forward | ACCATCCTGTGACTCTTGCG |
| | Reverse | TTCACCCACATCAGGCACTC |
| ADAMTS-4 | Forward | ACCGATTACCAGCCTTTGGG |
| | Reverse | CCGACTCCGGATCTCCATTG |
| ADAMTS-5 | Forward | CCGAACGAGTTTACGGGGAT |
| | Reverse | TGTGCGTCGCCTAGAACTAC |
| iNOS | Forward | ACACAGTGTCGCTGGTTTGA |
| | Reverse | AGAAACTTCCAGGGGCAAGC |
| Cox-2 | Forward | ATCAGAACCGCATTGCCTCT |
| | Reverse | GCCAGCAATCTGTCTGGTGA |
| CCL3 | Forward | TGCCAAGTAGCCACATCCAG |
| | Reverse | CACAGTGTGAGCAACTGGGA |
| CCL2 | Forward | TAGCATCCACGTGCTGTCTC |
| | Reverse | CAGCCGACTCATTGGGATCA |
| Aggrecan | Forward | CAGATGGCACCCTCCGATAC |
| | Reverse | GACACACCTCGGAAGCAGAA |
| Collagen II | Forward | GGCCAGGATGCCCGAAAATTA |
| | Reverse | ACCCCTCTCTCCCTTGTCAC |
| $\beta$-actin | Forward | AACCTTCTTGCAGCTCCTCCG |
| | Reverse | CCATACCCACCATCACACCCT |

## Luciferase assay

In addition to western blotting, we also used a luciferase assay to investigate the effect of cordycepin on NF-$\kappa$B activity. One day before transfection, NP cells were transferred to a 96-well plate at a density of $5 \times 10^3$ cells/well. NP cells were cotransfected with 20 ng of NF-$\kappa$B luciferase reporter and 5 ng of pRL-TK plasmids for each well. Lipofectamine 2000 (Invitrogen) was used as a transfection reagent. After transfection, the cells were cultured for 24 h and then pretreated with cordycepin (10, 50 or 100 µM) and/or stimulated with 10 µg/ml LPS for 24 h. Then, the cells were harvested for the luciferase assay. A Dual-Luciferase Reporter Assay System (Promega) was used for firefly and renilla luciferase activity measurements.

## Organ culture

Sprague Dawley rats (6–8 weeks old, mixed male and female) were euthanized. Then, the motion segment was isolated from each lumbar vertebra, including the upper and lower end plate and the whole disc. Intervertebral discs were cultured in a 24-well plate with 1 ml of complete DMEM medium (high-glucose DMEM with 10% FBS, 100 U/ml penicillin and 100 µg/ml streptomycin). Additional NaCl was added to the medium to increase the osmolarity to 410 mOsm/kg. Then, 100 µmol (10 µl) cordycepin (cordycepin group) or the same volume of saline (control and LPS groups) was injected into each disc through the annulus fibrosus using a microsyringe (5 µl, Hamilton). Then, the IVD was maintained

for 7 days with or without 10 µg/ml LPS and 100 µM cordycepin. The culture medium was replaced daily. A total of 30 Sprague Dawley rats, including 90 lumbar segments, were used in the organ culture (30 lumbar segments for each group).

### Dimethylmethylene blue assay

A DMMB assay was used to measure the proteoglycan (PG) content during 7 days of IVD organ culture. At specific time points, NP tissue was isolated from each cultured IVD and then digested with 5 mg/ml papain (sigma) prior to the DMMB assay. The amount of PG was normalized by the Total DNA of the NP tissue. The total DNA content was determined via an assay of total DNA using the PicoGreen kit (Molecular Probes). The DMMB assay was performed as previously described using chondroitin sulfate as a standard (*Li et al., 2015*). The PG level of each sample was expressed as µg PG /60 ng DNA (equal to $10^4$ cells).

### Histological analysis

Discs were removed from the culture medium after 7 days and fixed in 4% paraformaldehyde. After fixation, the discs were decalcified in EDTA for 14 days. Serial sagittal sections of discs (5-µm thick) were obtained to prepare slides. NBT/DAPI staining was used to evaluate the cell viability as previously described (*Lim et al., 2006*). Live cells were defined as cells with both DAPI and NBT staining, and cells with DAPI staining alone were registered as dead. The live/dead cell ratio was calculated using fluorescence microscopy (OLYMPUS) at 400× magnification. We analyzed three sections from each IVD tissue, and for each section, we calculated three fields and took the average. The cell viability was calculated as follows: (cell viability = live cells in field/total cells in field) × 100%.

Sagittal sections were also stained with hematoxylin and eosin (HE) and Safranin O-fast green to assess the degeneration of the IVD. Type II collagen and Aggrecan expression was detected using mouse monoclonal antibodies (1:200; Abcam) and a horseradish peroxidase-conjugated anti-mouse antibody (1:100; Dako), followed by color development with diaminobenzidine tetrahydrochloride (DAB, Dako). The results of the type II collagen and aggrecan staining were quantified as the IOD using the Image-Pro Plus 6.0 software. Cell and immunohistochemical staining was performed following standard histochemical protocols.

### Statistical analysis

All of the experiments were repeated 3 times. The data are expressed as the mean ± SD. A statistical analysis was performed with a one-way analysis of variance (ANOVA), followed by Duncan's post hoc test using SPSS 19.0 (IBM, Inc.). A *P*-value of less than 0.05 was considered statistically significant.

## RESULTS

### Cell Viability assay in the cell culture model

To study the potential cytotoxicity of cordycepin, we cultured NP cells with different concentrations of cordycepin for 24 h. As shown in Fig. 1, cordycepin did not show any cytotoxicity at concentrations of 10–100 µM ($P > 0.05$).

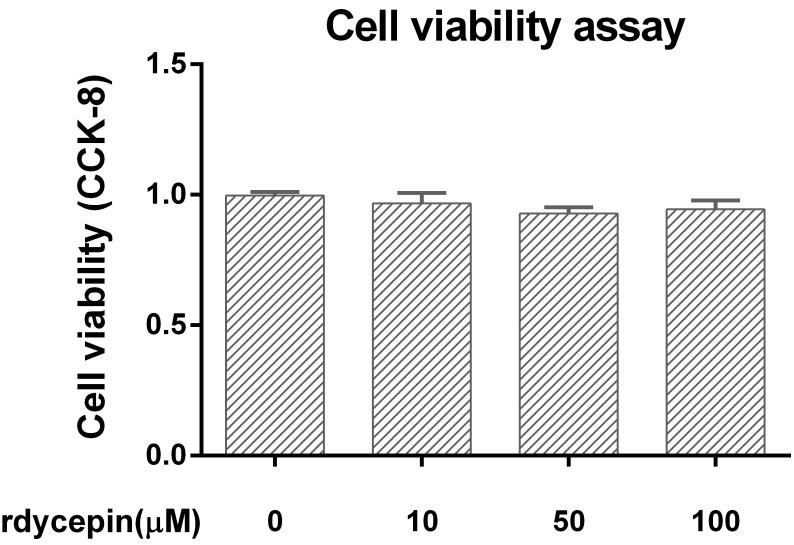

**Figure 1  Cell viability assay in the NP cell culture model.** We used CCK-8 to measure the NP cell viability in a monolayer culture model. Cordycepin did not show any cytotoxicity at concentrations of 10–100 µM.

## Cordycepin regulates LPS-induced matrix-degrading enzymes and extracellular matrix-related gene expression in NP cells

Monolayer cultures of NP cells were stimulated with 10 µg/ml LPS and 0, 10, 50 or 100 µM cordycepin, followed by a PCR assay and ELISA to measure the mRNA and protein levels respectively of various matrix-degrading enzymes. Cordycepin markedly inhibited the mRNA expression of multiple MMPs (MMP-3 and MMP-13) and ADAMTSs (ADAMTS-4 and ADAMTS-5) in a concentration-dependent manner (Figs. 2A–2D). Cordycepin also counteracted the LPS-induced gene downregulation of collagen-2 and aggrecan especially at a concentration of 50 or 100 µM. (Figs. 2E and 2F). The ELISA results showed that 50 and 100 µM cordycepin significantly inhibited MMP-3 and MMP-13 protein production (Figs. 2G and 2H). Moreover, cordycepin also reversed the LPS-induced increased in the gene expression of Cox-2 and iNOS at a concentration of 100 µM (Figs. 3A and 3B). Thus, cordycepin significantly inhibited PGE2 and NO production as induced by LPS in NP cells at a concentration of 100 µM (Figs. 3C and 3D).

## Cordycepin protects NP cells from LPS-induced matrix degradation

Given that cordycepin effectively suppressed multiple matrix-degrading enzymes that were upregulated by LPS, we further analyzed the effect of cordycepin on antagonizing the LPS-induced matrix degradation in NP cells. NP cells were stimulated with 10 µg/ml LPS in the presence or absence of cordycepin (100 µM) for 5 days, followed by collagen-II fluorescence staining and a western blot assay. These results indicate that the stimulation of NP cells with LPS strikingly reduced the collagen-II and aggrecan content, while cordycepin significantly inhibited the decrease of collagen-II and aggrecan compared to that observed in LPS group (Figs. 4A–4D).

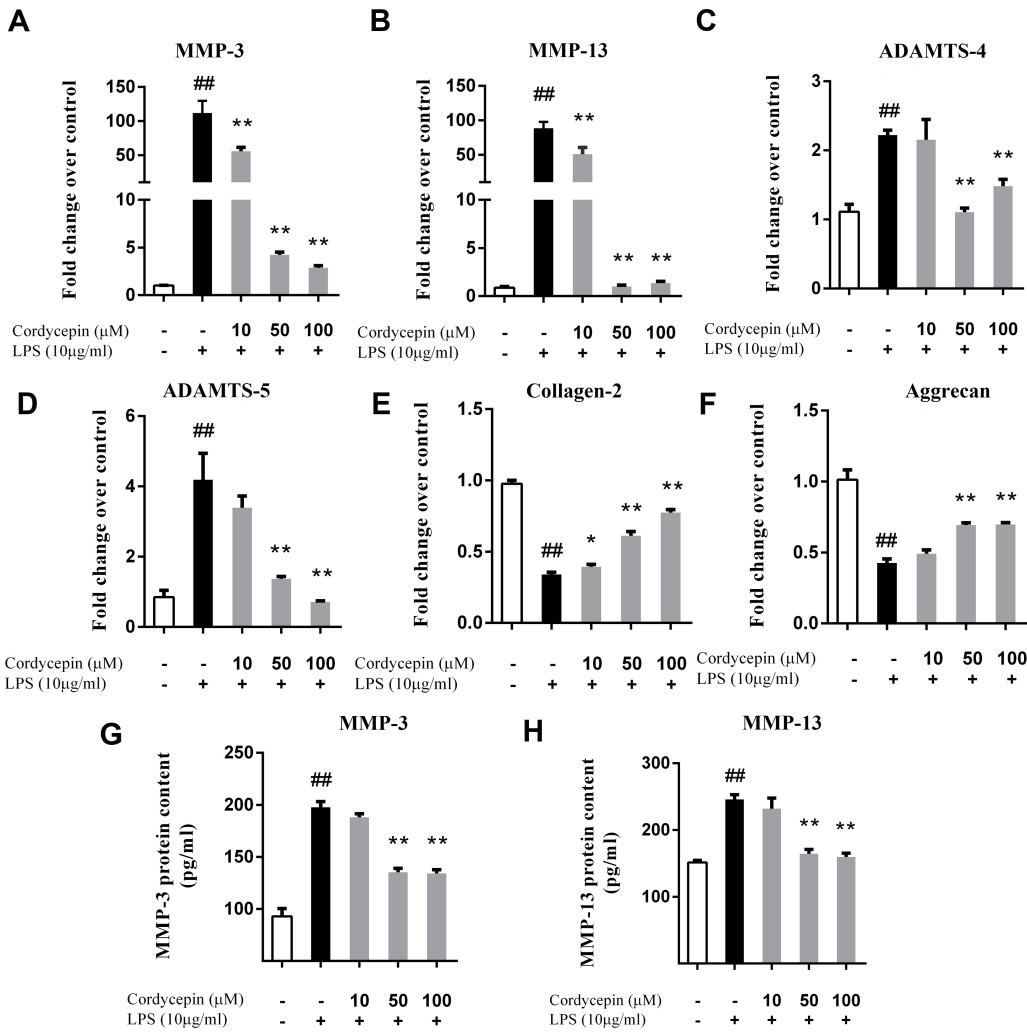

**Figure 2** **Cordycepin regulates the LPS-induced matrix-degrading enzymes and extracellular matrix related gene expression in NP cells.** We used PCR (A–F) and an ELISA assay (G, H) to investigate the effect of cordycepin on the LPS-induced gene expression and matrix-degrading enzyme secretion of NP cells. (A–F) Real-time PCR indicated that cordycepin downregulated the LPS-induced gene over-expression of MMP-3, MMP-13, ADAMTS-4 and ADAMTS-5. Moreover, cordycepin also counteracted the LPS-induced gene downregulation of collagen-2 and aggrecan especially at concentration of 50 mM or 100 $\mu$M. (G, H) An ELISA assay of MMP-3 and MMP-13 demonstrated that cordycepin inhibited LPS-induced MMPs secretion. The values are presented as the mean $\pm$ standard deviation. $^*P < 0.05$ compared to the LPS group; $^{**}P < 0.01$ compared to the LPS group; $^{\#\#}P < 0.01$ compared to the control group.

## Cordycepin inhibits the NP-mediated migration of macrophages after treatment with LPS

We examined the effect of NP cells that were treated with LPS on the migration of RAW 264.7 macrophages. Figure 5A shows that after treatment with 10 $\mu$g/ml LPS, NP cells promoted the chemotactic migration of macrophages compared to untreated NP cells. However, cordycepin inhibited the migration of macrophages promoted by NP cells after treating with LPS (Figs. 5A and 5B). Moreover, as shown in Fig. 5C–5E, 100 $\mu$M cordycepin

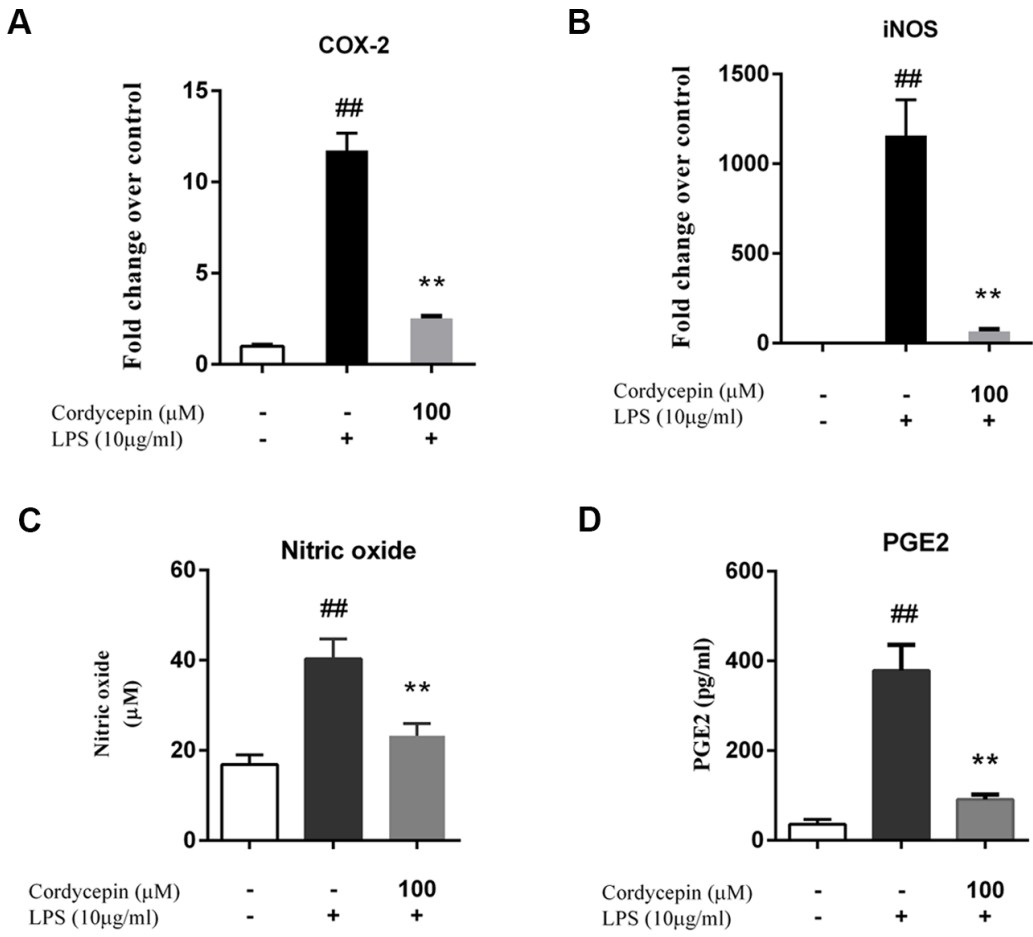

**Figure 3 Cordycepin decreases the LPS-induced production of PGE2 and NO in NP cells.** (A, B) Real-time PCR indicated that cordycepin reversed the LPS-induced increased gene expression of Cox-2 and iNOS. (C) The NO content was measured using the Griess reaction; the results showed that cordycepin inhibited LPS-induced NO production in NP cells. (D) An ELISA assay demonstrated that cordycepin inhibited the LPS-induced PGE2 production in NP cells. The values are presented as the mean ± standard deviation. **$P < 0.01$ compared to the LPS group; ##$P < 0.01$ compared to the control group.

reversed the LPS-induced gene upregulation of chemokine ligand 2 (CCL2, or monocyte chemotactic protein-1 (MCP-1)), a well-characterized macrophage chemotactic factor that is expressed in NP cells (*Yoshida et al., 2002*).

## Cordycepin inhibits the LPS-induced activation of the NF-$\kappa$B pathway in NP cells

To evaluate the potential involvement of signal transduction pathways and the mechanisms of the effects of cordycepin on LPS-stimulated NP cells, we measured the activation of the MAPK and NF-$\kappa$B pathways via western blotting. The results show that 10 $\mu$g/ml LPS significantly activated the MAPK and NF-$\kappa$B pathways in NP cells, while cordycepin inhibited the phosphorylation of I$\kappa$B$\alpha$ and p65 in a dose-dependent manner (Figs. 6A, 6D and 6E). However, cordycepin did not influence the activation of the MAPK pathway

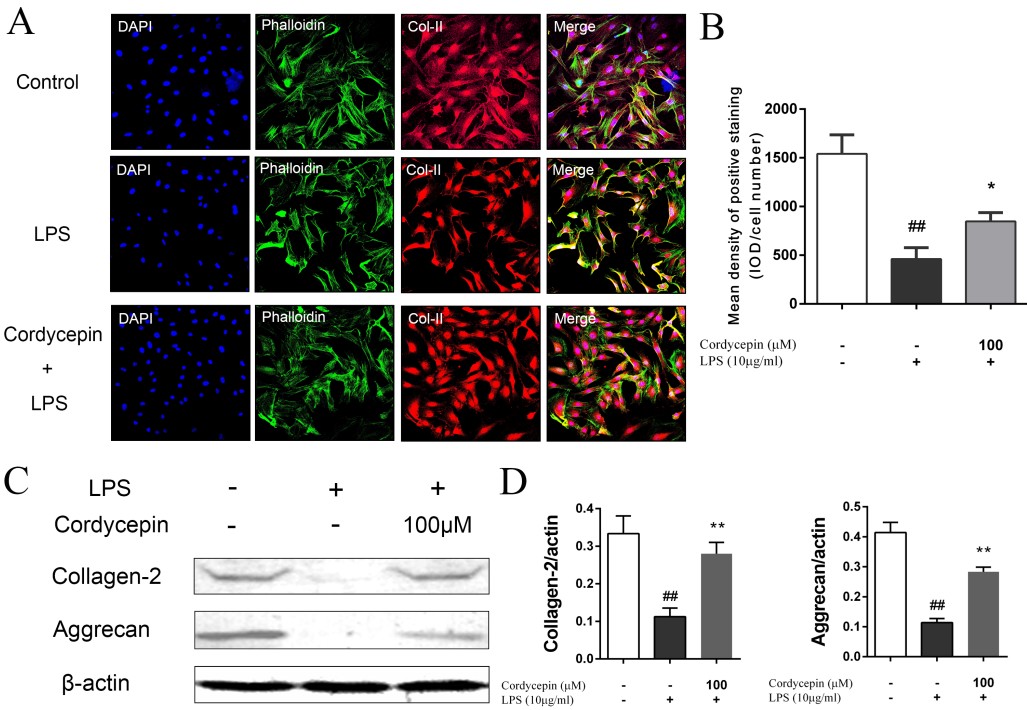

**Figure 4** **Cordycepin reverses LPS-induced matrix degradation in NP cells.** NP cells were treated with or without 100 μM cordycepin and 10 μg/ml LPS for 5 days and then fixed with 4% paraformaldehyde and used to perform fluorescence staining. (A, B) Collagen-II (Col-II) immunofluorescent staining showed that cordycepin reduced the LPS-induced collagen-II decrease in NP cells. (C, D) Western blotting results showed that cordycepin significantly reversed LPS-induced collagen-II and aggrecan loss. These results indicated that cordycepin could protect NP cells from LPS-induced matrix degradation. *$P < 0.05$ compared to the LPS group; ##$P < 0.01$ compared to the control group.

(Figs. 6B, 6F–6H). A luciferase assay also showed that cordycepin could inhibit the LPS-induced activation of the NF-$\kappa$B pathway in NP cells (Fig. 6C).

## Cordycepin reverses the LPS-induced degeneration of IVDs in an organ culture model

The NBT/DAPI staining showed that the cell viability of all groups at seven days was still greater than 80%, confirming the reliability of the organ culture model (Figs. 7A and 7B).

Figure 8A shows cultured IVD sections that were stained with HE and Safranin-O fast green. On the Safranin-O fast green sections, proteoglycans (PGs) stained red, and on the HE sections, they stained purple. After seven days of IVD organ culture, the presence of LPS resulted in severe PG loss compared to the effects of co-incubation with cordycepin (Fig. 8A).

The immunohistochemical staining of collagen-II and aggrecan were markedly enhanced in the NP area. Immunohistochemical staining showed that cordycepin effectively reversed the decreased collagen-II and aggrecan expression induced by LPS (Fig. 8B). The quantification of the Integral Optical Density (IOD) also indicates that the cordycepin group gained more collagen-II and aggrecan staining at day 7 (Fig. 8C, $P < 0.01$).

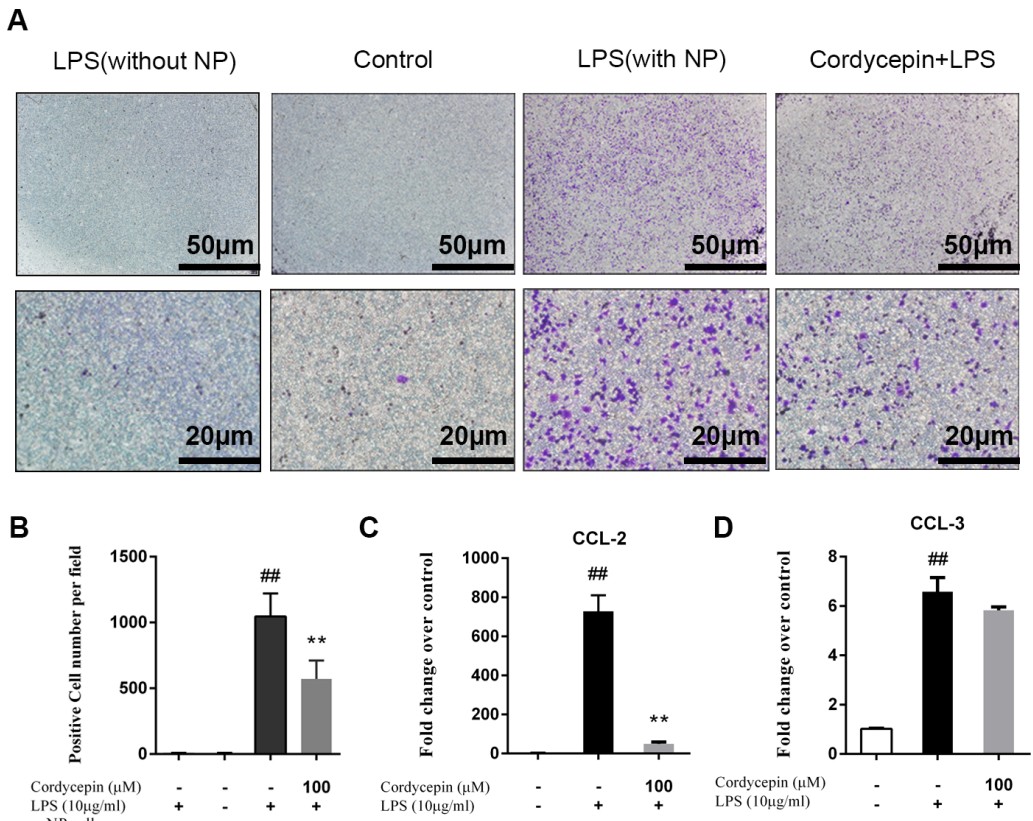

**Figure 5** **Cordycepin inhibits LPS-induced CCL2 expression and macrophage migration in NP cells.**
Macrophage migration was measured using a 24-well cell culture insert system. The results of crystal violet staining (A) and a positive cell count (B) showed that LPS promoted NP-mediated macrophage migration, which was inhibited by cordycepin. (C, D) Real-time PCR indicated that cordycepin inhibited the LPS-induced increased gene expression of CCL2 but not CCL3. The values are presented as the mean $\pm$ standard deviation. $^{*}P < 0.05$ compared to the LPS group; $^{**}P < 0.01$ compared to the LPS group; $^{\#\#}P < 0.01$ compared to the control group.

A DMMB assay was used to quantify the PG content of NP in an IVD culture. Similar to the histological results, cordycepin significantly attenuated the PG loss induced by LPS (Fig. 8D, $P < 0.01$).

## DISCUSSION

The present study provides, for the first time, evidence that cordycepin exhibits pharmacological anti-inflammation and anti-degeneration effects in LPS-induced NP cells and IVDs. Our results also show that cordycepin blocks the LPS-induced activation of the NF-$\kappa$B pathway, but not the MAPK pathway in NP cells.

The expression of MMPs and ADAMTSs has been extensively studied in the process of IVD degeneration. According to many studies (Vo et al., 2013; Weiler et al., 2002; Le Maitre, Freemont & Hoyland, 2004), MMP-3 and MMP-13 are not expressed in nondegenerated human discs but have increased expression in degenerated human discs. In addition, MMP-3 protein expression has a positive correlation with IVD histomorphological degenerative

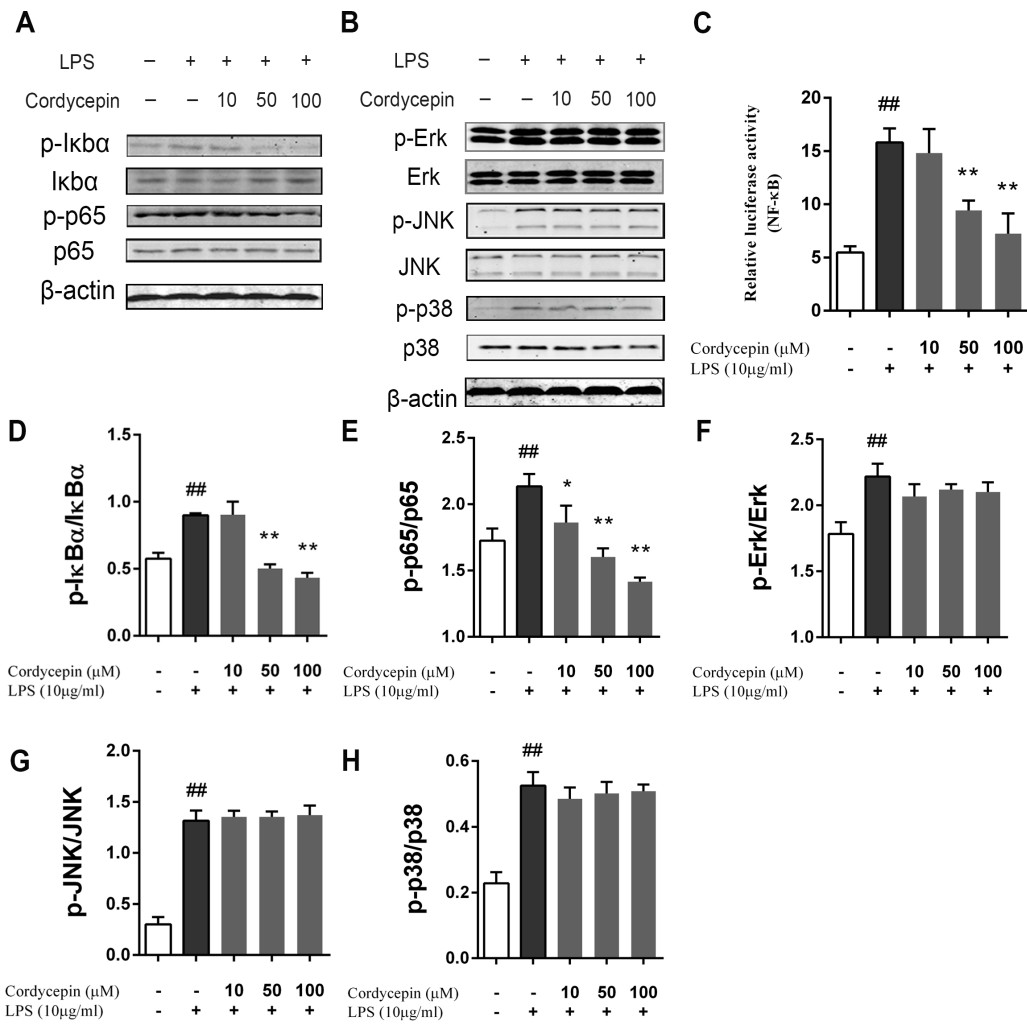

**Figure 6** **Effect of cordycepin on the LPS-induced activation of the NF-κB and MAPK pathways.** NP cells were pretreated with various concentrations of cordycepin for 2 h and then stimulated with 10 μg/ml LPS for 24 h. Then, western blotting was performed to evaluate the mechanism of cordycepin on LPS-treated NP cells. (A) Cordycepin significantly inhibied the phosphorylation of IκBα and p65 induced by LPS. (B) Cordycepin did not influence the phosphorylation of ERK, p38 or JNK enhanced by LPS. (C) NP cells were transfected with a NF-κB luciferase reporter, and the NF-κB pathway activity was determined by luciferase assay using a commercially available kit. (D–H) Quantitative analysis of western blotting data show cordycepin significantly inhibited the activation of the NF-κB pathway induced by LPS at concentrations of 50 and 100 μM. The values are presented as the mean ± standard deviation. $*P < 0.05$ compared to the LPS group; $**P < 0.01$ compared to the LPS group; $\#\#P < 0.01$ compared to the control group.

findings (*Weiler et al., 2002*). The upregulation of ADAMTSs has also been observed in degenerative discs (*Pockert, Richardson & Le Maitre, 2009*). ADAMTS-4 and ADAMTS-5 have been reported as the most important aggrecanases due to their strong capabilities in cleaving aggrecan among the 20 different ADAMTSs (*Tortorella et al., 1999*; *Gendron et al., 2007*). Inhibitors of MMPs and ADAMTSs have a therapeutic effect on OA (*Gilbert et al., 2007*) and IDD (*Leckie et al., 2012*) *in vitro* and *in vivo*. Collagen-II and aggrecan are the main components of nucleus pulposus, and the reduction of collagen-II content is highly

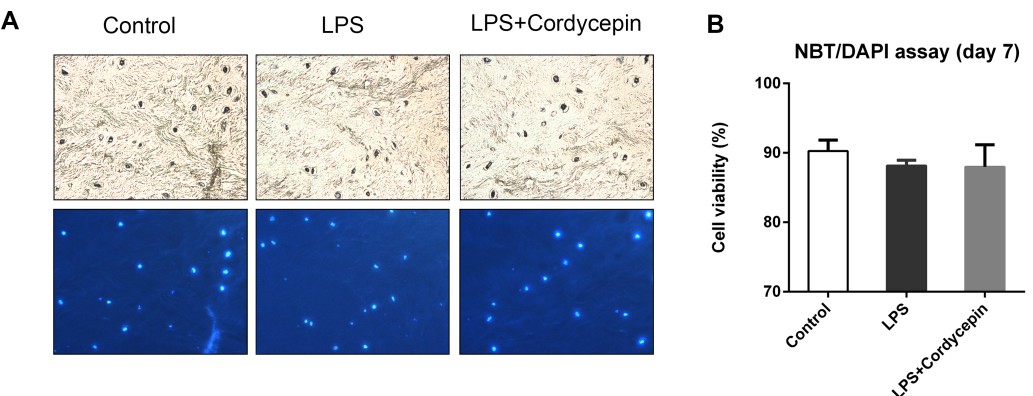

**Figure 7 Cell viability assay in the IVD organ culture model.** (A) NBT/DAPI staining on the 7th day of IVD organ culture. (B) Cell viability of the organ culture model. Cell viability = (NBT positive cell number/DAPI positive cell number) ×100%.

correlated with disc degeneration (*Shi et al., 2015*; *Kozaci et al., 2006*). Our study shows that cordycepin can both prevent LPS-induced collagen-II and aggrecan loss and promote their synthesis.

Many studies have revealed that NO and PGE2 play important roles in both the regulation of cellular metabolism in discs under mechanical stress conditions and the pathology of intervertebral disc degeneration (*Wang et al., 2011*; *Takada et al., 2012*; *Hou et al., 2014*). According to previous reports, PGE2 increases the excitability of rat sensory neurons and is involved in the development of sciatica in herniated disc disease (*England, Bevan & Docherty, 1996*). In addition, NO contributes to the development of radiculopathy by mediating protein nitration, moreover, the symptoms can be relieved by its suppression (*Lee et al., 2013*). Macrophages also play a critical role in the inflammatory response that is associated with degenerative disc disease and LBP (*Gawri et al., 2014*; *Gruber et al., 2015*). Macrophages could upregulate many cytokines expression including PGE2, and the interaction of macrophages and IVDs induce MMP-3 production which may cause extracellular matrix resorption (*Haro et al., 2000*). Previous studies have suggested that chemokine ligand 2 (CCL2) and chemokine ligand 3 (CCL3) are important mediators of macrophage infiltration into disc tissue (*Gawri et al., 2014*; *Gruber et al., 2015*). In our research, LPS upregulated the gene expression of CCL2 and CCL3 in NP cells, thereby promoting macrophage migration in NP cells and in a RAW 264.7 cell co-culture model. However, cordycepin reversed LPS-induced CCL2 gene upregulation of NP cells and inhibit macrophage migration, thus reducing the early-stage inflammation of NP cells. We also demonstrated that cordycepin could inhibit LPS-induced iNOS and Cox-2 gene overexpression and reduce the production of NO and PGE2 in NP cells.

The MAPK and NF-$\kappa$B pathways play important roles in the regulation of the inflammatory response (*Tak & Firestein, 2001*; *Berenbaum, 2004*). According to previous studies, cordycepin exerts anti-inflammatory effect via suppressing the NF-$\kappa$B pathways in many types of cells (*Jeong et al., 2010*; *Ren et al., 2012*; *Kim et al., 2011*). Cordycepin has also been reported to induce apoptosis in various cancer cells via MAPK pathway

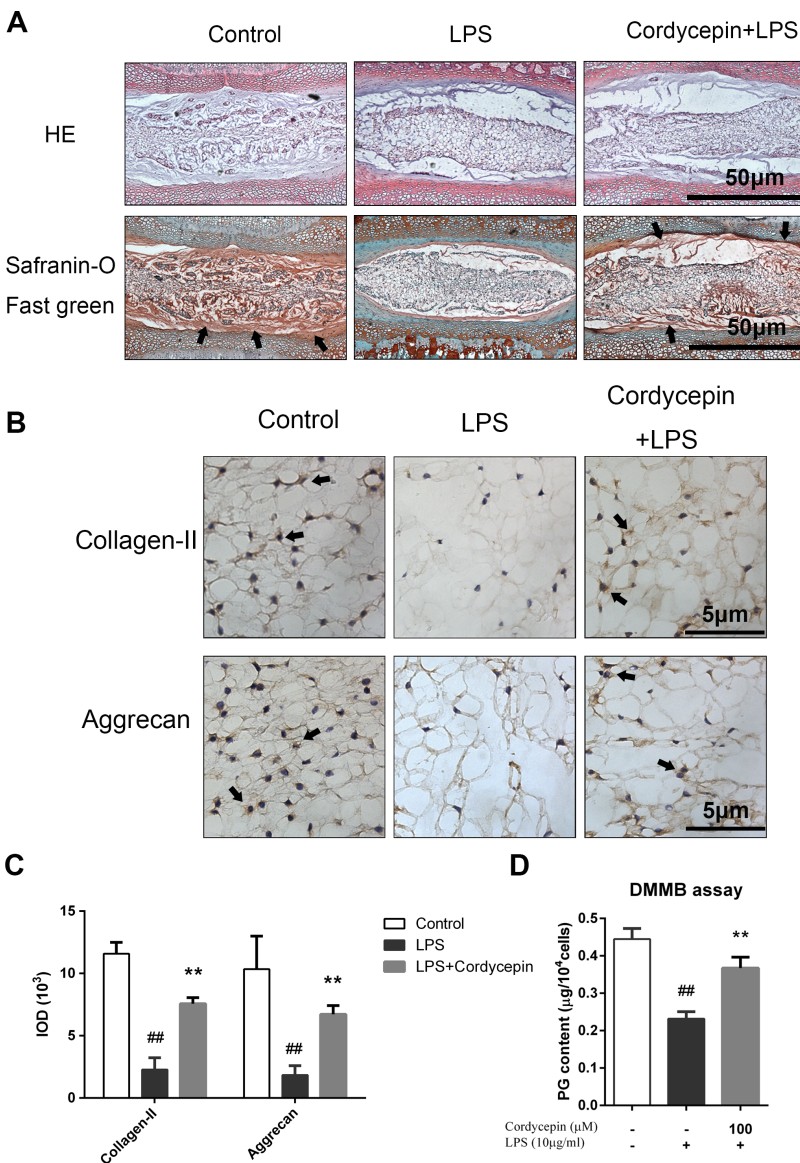

**Figure 8 Histological assessments and DMMB assay.** The lumbar spine discs of rat were cultured with or without LPS and cordycepin for seven days. Then, the discs were divided into two groups: one group was fixed and used to prepare serial section slides for staining. Fresh nucleus pulposus tissue was removed from another group of discs for DMMB assay. (A) HE and Safranin O-fast green staining. (B) Collagen II and Aggrecan immunohistochemistry staining. (C) The results of immunohistochemical staining were quantified in integral optical density (IOD). (D) A DMMB assay was used to quantify the PG content of NP in an IVD culture. Both chemical staining and immunohistochemical staining showed that cordycepin protected the intervertebral disc from LPS-induced PG loss. The black arrow shows the PG content in Safranin O-fast green staining and positive immunohistochemical staining. The values are presented as the mean ± standard deviation. $**P < 0.01$ compared to the LPS group; $##P < 0.01$ compared to the control group.

(*He et al., 2010*). In our study, cordycepin inhibited the NF-$\kappa$B pathway by inhibiting the phosphorylation of I$\kappa$B$\alpha$ and p65 in LPS-stimulated NP cells. In contrast, the phosphorylation of Erk, JNK and p38 of the MAPK pathway did not significantly change. This result was consistent to the data reported previously (*Jeong et al., 2010*; *Ren et al., 2012*).

Given that the *in vitro* experiment showed encouraging results, we further study the potential therapeutic effects of cordycepin in an organ culture model. Although the permeability of the annulus fibrous and endplate is very limited, our previous and other studies have confirmed that LPS and bioactive molecules can penetrate into the nucleus pulposus and exert bioactive effects (*Li et al., 2015*; *Kim et al., 2013*). Our results clearly indicate that high concentrations of LPS (10 $\mu$g/ml) can induce the degeneration of cultured IVDs in a relatively short period (seven days), while cordycepin can significantly reduce the LPS-induced PG loss in cultured IVD. A similar study reported that cordycepin may suppress IL-1$\beta$-stimulated catabolic enzyme, COX-2 and iNOS gene expression in chondrocytes, exerting chondroprotective effect and interfering with the inflammatory response (*Hu et al., 2014*). Considering the similarity between chondrocytes and NP cells, this previous study support our results to some extent. However, we focus more on the early stage of inflammation of NP cells in the present study, while the chondrocytes used in the previous study were from subjects with advanced-stage osteoarthritis.

Previous study have reported fibrin-genipin annulus fibrosus sealant as a drug delivery system for IDD, and it can maintain the bioactivity of the drug for over 20 days (*Likhitpanichkul et al., 2015*). Based on our study, cordycepin needs to be directly deliver to IVDs before exerting its anti-degenerative effects. Future investigations are required to determine if such a system can effectively deliver cordycepin and allow it to exert its effects in IVDs.

In conclusion, our results show that cordycepin exhibits a strong anti-inflammatory and anti-catabolic effect by inhibiting LPS-induced NF-$\kappa$B activation in NP cells. Our organ culture model also demonstrates an anti-degeneration effect of cordycepin in IVDs. Cordycepin may be a potential new agent for treating IDD in the future.

## ACKNOWLEDGEMENTS

The authors thank the staff of the Shanghai Key Laboratory of Orthopedic Implants.

### Funding

This work was supported by grants from the National Natural Science Foundation of China (81272038). The funders had no role in study design, data collection and analysis, decision to publish, or preparation of the manuscript.

### Grant Disclosures

The following grant information was disclosed by the authors:
National Natural Science Foundation of China: 81272038.

## Competing Interests

The authors declare there are no competing interests.

## Author Contributions

- Yan Li conceived and designed the experiments, performed the experiments, analyzed the data, contributed reagents/materials/analysis tools, wrote the paper, prepared figures and/or tables, reviewed drafts of the paper.
- Kang Li performed the experiments, analyzed the data, contributed reagents/materials/analysis tools, wrote the paper, prepared figures and/or tables, reviewed drafts of the paper.
- Lu Mao contributed reagents/materials/analysis tools.
- Xiuguo Han performed the experiments, analyzed the data, contributed reagents/materials/analysis tools, prepared figures and/or tables.
- Kai Zhang performed the experiments, contributed reagents/materials/analysis tools, wrote the paper.
- Changqing Zhao wrote the paper, prepared figures and/or tables.
- Jie Zhao conceived and designed the experiments, reviewed drafts of the paper.

## Animal Ethics

The following information was supplied relating to ethical approvals (i.e., approving body and any reference numbers):

All of the animal work was conducted according to relevant national and international guidelines and was approved by the Animal Experimental Ethical Committee of Shanghai Ninth People's Hospital (Approval number: 2013-47).

## Data Availability

Figshare: https://figshare.com/s/53cc3745ecacea5242d5.

DOI 10.6084/m9.figshare.2056077.

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
