# Peer review of "Cordycepin inhibits LPS-induced inflammatory and matrix degradation in the intervertebral disc"

_PeerJ, doi:10.7717/peerj.1992_

## Round 0.1 · original submission · Major Revisions

Dear authors,

A major revision of your manuscript is required before it can be reconsidered for publication.Please make a point to point reply to each of the reviewers comments when you submit your revised manuscript. Highlight the parts which have been revised and refer to these modifications in your response letter.

Reviewer 1 ·

Basic reporting

No comments

Experimental design

No comments

Validity of the findings

No comments

Additional comments

The manuscript titled “Cordycepin inhibits LPS-induced inflammatory and matrix degradation in the intervertebral disc” by Li Yan et al. examined the protective effects of cordycepin against LPS induced inflammation using primary cell as well as organ culture. The results are interesting. However I have several concerns that need to be addressed:
1. First of all the quality of English needs to be improved. For example; 1st line in abstract “cordycepin is an extract of the active phenolic component.....” needs modification to something like “cordycepin is a component of the extract obtained from....’’ because it is a purified compound of the extract not extract. Similarly in many other parts of the paper the language is not clear and correct; e.g. In Introduction, line 18 and 21. Heading “Histologic analysis” should be written as “Histological…” and similarly on page 15, line 16. Page 15, line 12 “IOD” stands for “Integral” Optical Density or “Integrated” Optical Density? In Discussion page 3: the statement “ Our preliminary results...” is not an appropriate statement. The preliminary result usually means that needs to be verified. If such is the case then why publish these results? Please correct language to avoid these misinterpretations.
2. The method of isolating NP cells is not clearly explained. Please give more details to provide reference if earlier published method has been followed. There can be contamination by other cells such as chondrocytes. Was it verified or not?
3. I also feel the dose used of both LPS and cordycepin on the higher side. Can the authors explain the choice of dose for both LPS and cordycepin. Many studies show the anti-inflammatory effects of cordycepin at doses lower than the used for present study.
4. The data shown in figure 4 and the text explaining this data in results section seems to be contradictory. Data show that LPS treatment reduced the staining which was further reduced by cordycepin treatment while text claim that cordycepin reversed the changed mediated by LPS. It need to be clarified.
5. The statistical analysis is presented for LPS vs LPS+ cordycepin groups. The statistical analysis between control and LPS group also needs to be conducted. Also, Figure 6 is a western blot data; one should avoid using the word ‘significant’ on data where statistical analysis has not been conducted. Some of the graphical results need to be explained in the text eg. Figure 5E about CCL3.
6. Also I felt some of the papers have not been cited by the authors. For example; Mol Med. 2011 Sep-Oct;17(9-10):893-900 is a paper which reported the effect of cordycepin on LPS induced inflammation by modulating NF-kB activation. Such articles need to be cited.

Reviewer 2 ·

Basic reporting

Abstract: Does not follow PeerJ format

Experimental design

No comments

Validity of the findings

No Comments

Additional comments

The manuscript “Cordycepin inhibits LPS induced inflammatory and matrix degradation in the intervertebral disc” investigates regulatory effects cordycepin has on inflammatory and catabolic mediators involved in disc degeneration by using both rat NP cells and an organ culture model. The study provides convincing evidence that cordycepin can attenuate LPS-mediated NP inflammation and thus adds to a growing body of literature concerning therapeutics for disc degeneration derived from natural products as well as potential uses of cordycepin. However, the present study has many issues that need to be addressed before it is suitable for publication.


Minor grammatical editing is required.
Introduction
The introduction is very brief and lacks sufficient detail to provide readers with the necessary background.
1. The authors outline the involvement of catabolic proteases in disc degeneration but do not introduce the role different inflammatory mediators have in degeneration. This is important back ground since the authors are using an inflammatory model. Refer to the following papers for more information
a. Hoyland et al 2008 Investigation of the role of IL-1 and TNF in matrix degradation in the intervertebral disc. Rheumatology
b. Phillips et al The cytokine and chemokine expression profile of nucleus pulposus cells: implications for degeneration and regeneration of the intervertebral disc. Arthritis res ther
c. Wuertz and Haglund 2013 Inflammatory Mediators in Intervertebral Disk Degeneration and Discogenic Pain. Global Spine Journal
d. Krock et al 2014 Painful, degenerating intervertebral discs up‐regulate neurite sprouting and CGRP through nociceptive factors. J Cell Mol Med
2. There is extensive research on cordycepin. The authors need to introduce its known and/or hypothesized mechanisms of action
3. Line 57-59 contains several grammatical errors. Additionally give specific examples of anti-inflammatories
4. The authors of the current study use an LPS model to mimic disc degeneration instead of IL-1B or TNF which are more commonly used. LPS is a TLR4 ligand and the role of TLRs in disc pathobiology is increasingly being thought to be important. Since the authors use LPS, it is necessary that they introduce TLRs and their possible role in disc degeneration and low back pain. Please refer to the papers below
a. Krock et al 2015 Nerve Growth Factor is Regulated by Toll-like Receptor 2 in Human Intervertebral Discs. J Biol Chem.
b. Quero et al 2013. Hyaluronic acid fragments enhance the inflammatory and catabolic response in human intervertebral disc cells through modulation of toll-like receptor 2 signalling pathways. Arthritis Res Ther.
c. Klawitter et al 2014. Expression and regulation of toll-like receptors (TLRs) in human intervertebral disc cells. Eur Spine J
d. Krock, Rosenzweig, Haglund 2015 The Inflammatory Milieu of the Degenerate Disc: Is Mesenchymal Stem Cell-based Therapy for Intervertebral Disc Repair a Feasible Approach? Curr Stem Cell Res Ther
e. Gawri et al 2014. High mechanical strain of primary intervertebral disc cells promotes secretion of inflammatory factors associated with disc degeneration and pain. Arthritis Res Ther
5. The authors look at macrophage migration in their experiments. Rational to why they did this must be included in the intro
Methods – several details must be specified
6. Include more detail about cell isolation procedure. Is this technique commonly used?
7. What antibiotics were used for culture?
8. Were cell and organ culture experiments done in the presence of serum or serum free media?
9. How was the concentration of LPS determined?
10. Was DMSO used in control media as a vehicle
11. Include reference for griess reaction/nitric oxide measurment
12. What liquids were used for immunofluorescence and western plot (ie TBS? PBS?)
13. What is the binding site of the p-p65 antibody used. Several site can be phosphorylated which can then effect function
14. How was RNA isolated prior to cDNA synthesis
15. ΔΔCt should read 2- ΔΔCt. This also requires a reference
16. Authors say they performed a BCA assay prior to western blot. How much protein was loaded into the gel.
17. Animals – specify sex.
a. Justify why such young animals were used when disc degeneration is a pathology that affects middle and middle-late age adults

Results
1. While the authors include gene expression and protein data of catabolic proteases and inflammatory mediators in cell culture experiments they do not look at anabolic factors related to matrix synthesis. Presumably LPS treatment negatively effects NP cells’ ability to produce matrix, therefore it is important to examine whether cordycepin restores matrix synthesis
a. The authors need to include gene expression data of aggrecan and col II
b. The authors also need to include protein data for aggrecan and Coll II, for example by western blot
2. Will RAW 264.7 cells migrate in the presence of LPS without NP cells? This seems like an important control to include.
3. How do the authors know that the CCL2 gene expression they are evaluating is from NP cells and not RAW cells?
4. The authors say that cordycepin inhibits LPS reduced Col II and demonstrate this by immunofluorescence in figure four. However, looking at the figure there appears to be little to no increase in Col-II staining in the cordycepin+LPS group compared to the LPS group alone. For this reason it is necessary for the authors to include other data showing col II levels. Furthermore aggrecan should be investigated since it is a principle component of the NP (see point above).
a. Furthermore, the authors include a graph quantifying relative fluorescence. This graph shows cordycepin decreases Col II compared to LPS. This goes directly against what the authors state in their figure legend and results.
5. Densitometry needs to be included for the signaling western blots in figure 6
6. Colour schemes/greyscale for graphs need to be standardized
7. Scale bars need to be included for all microscopy pictures
8. For signaling western blots the area cropped out should be reduced in order to show more of the membrane above and below the bands currently shown in the figure. It appears in some blots, such as IkBa, that there may be other bands at the crop boundary.
9. Figure 8 – in the results the authors mention NP and AF tissues show increased col II and aggrecan staining with cordycepin. However in figure 8 they only show a single set of images (resumably of one cell type). This is not defined in the figure legend. Please fix and include the other tissue type. If both tissues are observed in the included micrographs, please define the area between NP and AF
Discussion
In general the authors need to make a greater effort to discuss their results and explain how they add to the bodies of literature concerning disc degeneration and cordycepin as a therapeutic. As is, the discussion is mainly restating the results
1. The authors describe the changes of col II they observed as degradation. However, it could be decreased synthesis, rather than degradation, since they did not examine Col II degradation products. Therefore it is only appropriate to say “Coll II decreases”
2. The authors should discuss cordycepin mechanisms and why it may only effect nf-kb and not other signaling pathways investigated.
3. How would cordycepin be delivered as a therapeutic to treat disc degeneration and back pain.
4. Similar studies to the present one have been carried out using cordycepin and chondrocytes. The authors of the current study should discuss these and relate their own results to them.

---

## Round 0.2 · accepted · Accept

The reviewers are satisfied with the revised version of your manuscript.

Reviewer 1 ·

Basic reporting

No cooments

Experimental design

No comments

Validity of the findings

No comments

Additional comments

The authors have addressed the queries. The manuscript is acceptable in present form.

Reviewer 2 ·

Basic reporting

Significantly improved

Experimental design

significantly improved

Validity of the findings

ok

Additional comments

no further comments